# Multi-Targeting Neuroprotective Effects of *Syzygium aromaticum* Bud Extracts and Their Key Phytocompounds against Neurodegenerative Diseases

**DOI:** 10.3390/ijms24098148

**Published:** 2023-05-02

**Authors:** Himadri Sharma, Dan Yeong Kim, Kyu Hwan Shim, Niti Sharma, Seong Soo A. An

**Affiliations:** Department of Bionano Technology, Gachon Bionano Research Institute, Gachon University, 1342 Seongnam-daero, Sujeong-gu, Seongnam-si 461-701, Gyeonggi-do, Republic of Korea

**Keywords:** Alzheimer’s disease, *Syzygium aromaticum*, H_2_O_2_ induced stress, anti-acetylcholine esterase activity, anti-glycation, MDS, oxidative stress

## Abstract

Alzheimer’s disease (AD) is a neurodegenerative disease that causes a gradual loss of normal motor and cognitive function. The complex AD pathophysiology involves various factors such as oxidative stress, neuroinflammation, amyloid-beta (Aβ) aggregation, disturbed neurotransmission, and apoptosis. The available drugs suffer from a range of side effects and are not able to cover different aspects of the disease. Therefore, finding a safer therapeutic approach that can affect multiple targets at a time is highly desirable. In the present study, the underlying neuroprotective mechanism of an important culinary spice, *Syzygium aromaticum* (Clove) extract, and major bioactive compounds were studied in hydrogen peroxide-induced oxidative stress in human neuroblastoma SH-SY5Y cell lines as a model. The extracts were subjected to GC-MS to identify important bioactive components. The extracts and key bio-actives reduced reactive oxygen species (ROS), restored mitochondrial membrane potential (MMP), and provided neuroprotection from H_2_O_2_-induced oxidative stress in cell-based assays due to the antioxidant action. They also reduced lipid peroxidation significantly and restored GSH content. Clove extracts have also displayed anti-acetylcholinesterase (AChE) activity, anti-glycation potential, and Aβ aggregation/fibrilization inhibition. The multitarget neuroprotective approach displayed by Clove makes it a potential candidate for AD drug development.

## 1. Introduction

Neurodegenerative diseases (NDDs) are a group of diseases affecting various regions of the brain and causing progressive deterioration of intellectual faculties such as learning, cognition, memory, and motor skills. Alzheimer’s disease (AD) is one of the most prevalent NDDs, affecting millions all over the world. The aggregation of amyloid-beta (Aβ) peptide, tau protein hyperphosphorylation, lower levels of acetylcholine, neurofibrillary tangles, and neuronal damage from oxidative stress are important characteristics of AD pathophysiology [1]. The oxidative stress further generates a vicious cycle by generating more reactive oxygen species (ROS), superoxide radicals (O_2_^•−^), peroxynitrite (NO_3_^−^), and other reactive nitrogen species (RNS), affecting enzymes, proteins, and ion channels [2,3].

The existing therapies for improving AD symptoms work on a specific target, for example, acetylcholine esterase inhibitors (Donepezil, Galantamine, and Rivastigmine), anti-amyloid antibodies (Aduhelm), and NMDA receptor antagonists (Memantine). Unfortunately, these drugs exhibit side effects ranging from dizziness to amyloid-related imaging defects [4]. However, as AD is a multifaceted disease, a multitarget neuroprotective approach is required to ameliorate the symptoms. Therefore, there is an urgent need to identify compounds with better efficacy and minimal side effects for the treatment of AD. The therapeutic role of plant extracts has been long-established, and numerous phytocompounds have been identified with multiple neuroprotective properties [4,5].

The dried bud of *Syzygium aromaticum* (Clove; Family Myrtaceae) is a notable culinary spice that has been used for ages in folk medicine for its antiviral [6], anti-inflammatory [7], anti-nociceptive [8], anti-oxidant [9], anti-allergic [10], anti-microbial [11], anti-cancer [12], hepatoprotective [13], and neuroprotective [14] properties. Recent studies indicated the importance of exercise and clove oil supplementation in reversing memory deficits, apoptosis, and maintaining mitochondrial homeostasis in the hippocampus region of the AD brain [15]. It also modulated the seizures and alleviated neuronal death in the CA1 region of the hippocampus [16].

One of the major factors in NDDs is the generation of reactive oxygen species (ROS). Hydrogen peroxide (H_2_O_2_)-induced stress in the cultured cells generates ROS and accelerates other associated pathogenic reactions such as inflammation, mitochondrial dysfunction, tau hyperphosphorylation, the generation of neurofibrillary tangles, and apoptosis [17,18]. Therefore, protection against oxidative stress is an important therapeutic strategy for NDDs.

The present study has been designed to study the effect of hexane and ethyl acetate extracts of clove bud and bioactive compounds on H_2_O_2_-induced oxidative stress in SH-SY5Y neuronal cells. Additionally, other aspects of neuroprotection were explored by studying acetylcholine esterase (AChE) inhibitory potential, anti-oligomerization and fibrillation activity, and the ability to inhibit the formation of advanced glycation end products (AGEs) to elucidate the mechanism of neuroprotection by clove extract.

## 2. Results

### 2.1. Phytochemical Estimation and Antioxidant Potential of Clove Extract

Total phenolics (TPC) and flavonoids (TFC) were estimated in the extracts using colorimetric assays. Phenols and flavonoids are secondary metabolites with an important role in growth, communication, and plant defense [19,20]. The phenolic content in CL–H (Clove–Hexane) and CL–EA (Clove–Ethyl Acetate) extracts was estimated to be 91.24 ± 0.21 mg GAE/g and 48.75 + 0.10 mg GAE/g, respectively. The flavonoid content in the CL–EA (10.55 + 0.59 mg QE/g) was significantly higher than that in the CL–H (1.15 + 0.09 mg QE/g). For the determination of the antioxidant potential of the extracts, DPPH, ABTS, and FRAP assays were conducted. In the DPPH assay, the percent radical scavenging activity observed was 72.3 + 1.19% and 61 + 0.47% for CL–H and CL–EA, respectively. The FRAP assay is an electron transfer (ET)-based method that measures the reduction of Fe^3+^ to Fe^2+^ in the presence of antioxidants. CL–H exhibited a better FRAP value (214.95 + 0.007 µM Fe^2+^/g) compared to CL–EA (68 + 0.003 µM Fe^2+^/g). The better antioxidant potential of CL–H compared to CL–EA can be correlated with the higher phenolic content of CL–H. A positive correlation between phenolic content and antioxidant potential has already been reported [21].

### 2.2. GC–MS Analysis

The GC–MS chromatogram of hexane and ethyl acetate extracts of *Syzygium aromaticum* identified the important bioactive compounds in each extract. GC–MS recorded a total of 28 and 14 peaks for CL–EA and CL–H, respectively, based on retention time, peak area (%), height (%), and mass spectral fragmentation compared to those of the known compounds in the NIST library (Appendix A). The main components identified in CL–EA were eugenol (33.98%), caryophyllene (16.12%), caryophyllene oxide (9.55%), and 2′,3′,4′-trimethoxy acetophenone (8.92%) (Appendix A). On the other hand, CL–H had eugenol (48.93%), acetyl eugenol (20.13%), and caryophyllene (18.69%) (Appendix A). The profile of bioactive components in CL–EA and CL–H depends on the polarity of the solvent, as the EA fraction is moderately polar while hexane is non-polar. As a result, non-polar compounds become extracted in the hexane fraction first as a major percentage of eugenol, acetyl eugenol, and caryophyllene compared to the EA fraction. However, the EA fraction is considered safer and more active compared to the hexane fraction [22].

Eugenol is a compound with versatile pharmacological properties such as antidiabetic, neuroprotective, anti-cancer, antioxidant, anti-inflammatory, and analgesic. It has also been generally recognized as safe (GRAS) by the World Health Organization (WHO) [23]. Both caryophyllene and caryophyllene oxide possess antioxidant, anticancer, and analgesic properties. They are also approved as flavoring agents by the Food and Drug Administration (FDA) and the European Food Safety Authority (EFSA) [24]. Eugenyl acetate possesses anti-AChE and antioxidant activity [25,26]. The neuroprotective potential of other smaller (%) components identified in the extract, such as Humulene, Copane [27], Anethol [28], Vanillin [29], and Cubebene [30], has already been reported. Meanwhile, 2′,3′,4′—trimethoxy acetophenone is a heterocyclic thioether derivative of pyrogallic acid, and the literature on its therapeutic action is dearth. Hence, out of curiosity, we selected this compound along with the major phytocompounds, eugenol and β-caryophyllene, for further biochemical and cellular investigations.

### 2.3. In Vitro Anti-AGE Activity of Clove Extract

A BSA–AGE fluorescence assay was used to evaluate the anti-glycation potential of Clove extracts and eugenol, β-caryophyllene, and 2′,3′,4′-trimethoxy acetophenone (TMA) identified by GC–MS in the extracts (Appendix A). For the study, D-glucose was chosen as it is the key glycating sugar compared to other sugars. CL–EA potentially inhibited the glycation with an IC_50_ value of 83.09 μg/mL, comparable to the positive control, aminoguanidine hydrochloride (IC_50_ 53.27 μg/mL). In a previous report, aminoguanidine displayed an IC_50_ of 59.2 μg/mL [31], which is in agreement with the value reported by us. CL–H also exhibited good anti-AGE activity (IC_50_ 274.2 μg/mL) (Figure 1)**.** Earlier, the hydroalcoholic, aqueous, and methanolic extracts of Clove inhibited AGE formation (88%, 91.5%, and 72.8%, respectively) in the BSA-glucose model [32,33,34]. Eugenol exhibited anti-AGE activity (IC_50_ 152 μg/mL), while β-caryophyllene and TMA did not show any significant inhibition. Similar results were obtained previously where eugenol exhibited anti-AGE properties (IC_50_ 10 mM) [35], while β-caryophyllene failed to do so [36].

### 2.4. In Vitro Acetylcholinesterase Inhibitory Activity

Acetylcholinesterase (AChE; E.C.3.1.1.7) is an acetylcholine (ACh) hydrolyzing enzyme with an important role in the cholinergic signaling pathway. Inhibition of AChE is desirable in the treatment of AD to maintain ACh levels. Therefore, the extracts were initially screened for anti-AChE activity at 100 μg/mL, where CL–H and CL–EA displayed 50.97% and 56.64% inhibition, respectively. Galantamine hydrobromide was used as an inhibitor control for the assay, and it exhibited 96.75% inhibition at 10 μg/mL. The IC_50_ values (half maximal inhibitory concentration) of the extracts, eugenol, β-caryophyllene, and the positive control have been shown in Figure 2.

The IC_50_ values of extracts and compounds were 345.1 μg/mL (CL-EA), 268.3 μg/mL (CL-H), 5.6 μg/mL (eugenol), and 113 μg/mL (β-caryophyllene). However, TMA did not show any significant anti-AChE activity (28% at 1 mg/mL). The IC_50_ value of galantamine hydrobromide was calculated as 1.02 μg/mL, similar to the previously reported value of 1.45 μg/mL [37]. The IC_50_ value for eugenol obtained in our experiment was comparable to the earlier reported value of 40.32 μM (i.e., 6.62 μg/mL) [38]. 

The mode of inhibition of CL-H, CL-EA, eugenol, and β-caryophyllene was analyzed by a Lineweaver-Burk plot. The kinetic parameter values attained by extracts and pure compounds have been summarized in Table 1. Competitive inhibition was observed for the tested samples (Figure 3), where Vmax remained unaffected (~3.9 μmol/min/mg) while Km increased (17.64, 12.28, 14.19, and 17.87 mM) for CL-H, CL-EA, eugenol, and β-caryophyllene, respectively. In a previous study, eugenol inhibited snails’ AChE by a competitive-non-competitive mechanism (mixed inhibition) [39].

### 2.5. Clove Extract Reduced Aβ Oligomerization and Fibrilization

The MDS was used to evaluate the Aβ oligomerization inhibition effect of the clove extract and the pure compounds. To confirm the oligomerization inhibition potential, the samples were reacted with Aβ_1–42_ for 0, 2, and 4 h (Figure 4A–C). MDS results were calculated based on each sample’s 0 hr value; hence, all sample signals were at 1.0 (Figure 4A). At 2 h, oligomerization reduction was significant in CL-EA (** *p* < 0.01) and TMA (* *p* < 0.05) (Figure 4B). Eugenol also decreased the oligomerization at 2 h, but the *p* value was non-significant. As the time of incubation progressed, CL-H also decreased oligomerization significantly (* *p* <0.05) as did CL-EA; however, more significant reduction was shown by TMA (** *p* <0.01) going from 2 h to 4 h (Figure 4C). Considering the overall Aβ_1–42_ oligomerization tendency at every time point, the area under the curve (AUC) was calculated. As a result, CL-EA (* *p* <0.05) and TMA (** *p* <0.01) showed a significant decrease in Aβ_1-42_ oligomerization tendency with TMA performing better than CL–EA (Figure 4D). The possible explanation could be that as AUC is calculated over a period of time (0-4 h), depending on the kinetics of each compound, the reaction time with Aβ is different. In other words, since each compound may have a different phase of inhibiting Aβ, the effect of inhibiting Aβ appears at different times. The compound can degrade with time, but this degradation cannot be monitored by MDS. Hence, to summarize the results, TMA and CL-EA are effective in inhibiting Aβ oligomerization compared to CL-H and β-caryophyllene, which slightly decreased the oligomerization without a significant *p*-value, and eugenol was like the negative control (Figure 4D). 

To access the Aβ fibrilization inhibition activity of the samples, a ThT assay was conducted. ThT is a fluorescent dye that interacts with the β-sheet of the fibrils; consequently, it is used to monitor Aβ fibril formation. The fluorescence is increased upon binding of ThT to the amyloid fibrils. The extracts and compounds were screened (100 μg/mL) for Aβ fibrilization inhibition using phenol red as a positive control. ThT fluorescence was reduced significantly in the extracts compared to the control (buffer + Aβ). Phenol red displayed 71.34 ± 2.9% inhibition (*** *p* < 0.001) at 50 μM similar to the previously reported value [40]. Both CL-H and CL-EA exhibited potent inhibitory activity (*** *p* < 0.001) with 73.49 ± 5.53% and 86.59 ± 0.38%, respectively. On the other hand, eugenol, β-caryophyllene, and TMA displayed lower inhibition (48.38 ± 0.33%, *** *p* < 0.001; 18.44 ± 3.54%, ** *p* < 0.01; 9.13 ± 0.38%, **p* < 0.05, respectively) compared to the extracts (Figure 5). To summarize the results, TMA, and CL-EA effectively reduced Aβ oligomerization, while CL-EA and CL-H inhibited fibrilization to a greater extent compared to others. Previously, the aqueous extract of clove inhibited amyloid fibril formation by 90% in a concentration-dependent manner [41].

### 2.6. Noncytotoxic Effect of Clove Extracts on the SH-SY5Y Cell Line

The cellular viability was estimated [42] in the neuroblastoma cell line (SH-SY5Y) after 24 h of treatment with different concentrations of the extracts (1, 10, 25, 50 μg/mL) using WST-8 dye. No cytotoxicity was observed up to 25 μg/mL for CL-H and CL-EA with cell viability over 95%. However, at 50 μg/mL, the cell viability decreased to 88.4% (* *p* < 0.05) and 86.7% (** *p* < 0.01) for CL-H and CL-EA, respectively (Figure 6). Therefore, subsequent cell culture experiments were conducted using concentrations below 50 μg/mL to minimize cell death due to the extract’s toxicity. Similar results were obtained in a previous study where no cytotoxicity was observed up to 20 μg/mL when treating SH-SY5Y cells with the clove essential oil [43]. The pure compounds were tested at lower concentrations (0.01, 0.1, 1, 10 μg/mL), where eugenol and TMA displayed significant (* *p* < 0.05) decreases in cell viability (89.9% and 87.2%, respectively) compared to the control at 10 μg/mL. While the cell viability was over 93% for β-caryophyllene. 

### 2.7. Clove Extracts Protected against H_2_O_2_-Induced Oxidative Stress

The neuroprotective effects of the extracts were assessed by H_2_O_2_-induced oxidative stress in the cells. In our experiment, H_2_O_2_ at 100 μM resulted in 50% cell survival after 6 h of treatment. Hence, this H_2_O_2_ concentration was used to induce stress in SH-SY5Y cells pre-treated with the extracts for 12 h. CL-H performed better in protecting the cells against oxidative damage and significantly (^##^
*p* < 0.01) increased the cell viability at 0.1–1 μg/mL and 10-30 μg/mL (^#^
*p* < 0.05) as compared to H_2_O_2_ control. CL-EA significantly (^#^
*p* < 0.05) increased the cell viability at 0.1-1 μg/mL as compared to H_2_O_2_ control, after which it almost remained constant. In our experiment, pre-treatment with eugenol at lower concentrations (0.01 and 0.1 μg/mL) did not show neuroprotection; hence, higher concentrations (2.5 and 5 μg/mL) were used to study the effect. On the other hand, as β-caryophyllene displayed significant (^###^
*p* < 0.001) neuroprotection at 1 μg/mL, lower concentrations (0.01 and 0.1 μg/mL) were also used. TMA provided better neuroprotection (^###^
*p* < 0.001) than the others, even at the lower concentrations (0.1 μg/mL) (Figure 7**)**. Hence, the neuroprotective effect shown by Clove extract is due to the major bioactive components. Additionally, the better neuroprotection displayed by CL-H could be due to a higher content of eugenol and eugenyl acetate [44] as compared to CL-EA. The neuroprotective mechanism was further explored by measuring ROS and MMP.

### 2.8. Clove Extracts Prevented H_2_O_2_-Induced ROS Generation

Oxidative stress is one of the important factors in the pathogenesis of AD. To access the intracellular reactive oxygen species (ROS) scavenging activity of the extracts, the SH-SY5Y cells were pre-treated with the extracts for 12 h, followed by 4 h exposure to H_2_O_2_ (100 μM). The ROS production in the cells was monitored by a fluorescent dye (H_2_DCFDA), which is oxidized to DCF by ROS. The SH-SY5Y cells treated with H_2_O_2_ generated 154.8 ± 2.5% ROS compared to the untreated cells (100%). Pre-treatment of the cells with extracts resulted in a dose-dependent decrease in ROS production. The CL-H was more effective and significant (131.7 ± 0.6% at 1 μg/mL, *p* < 0.01; 122.5 ± 6.5% at 10 μg/mL; and 115.7 ± 2.5% at 25 μg/mL) in reducing ROS compared to CL-EA (150.7 ± 7.4% at 1 μg/mL; 134.5 ± 0.7% at 10 μg/mL; and 140.8 ± 2.5% at 25 μg/mL) Figure 8. Similarly, both eugenol and β-caryophyllene attenuated increased ROS levels significantly (^##^
*p* < 0.01) at 0.01 μg/mL and 0.03–0.3 μg/mL (^###^
*p* < 0.001), whereas TMA showed a significant dose-dependent decrease in ROS at 0.1–5 μg/mL (^###^
*p* < 0.001) (Figure 8). A possible explanation for better ROS scavenging activity by CL-H could be due to the higher content of eugenol, acetyl eugenol, and β-caryophyllene.

### 2.9. Restoration of Mitochondrial Membrane Potential by Clove Extract

Mitochondrial membrane potential (∆Ψm) is reduced due to oxidative damage to the cell. In the present study, H_2_O_2_ was used to induce oxidative stress in SH-SY5Y cells. The H_2_O_2_ concentration and time of induction were optimized using a fluorescent dye (TMRE), which has an affinity for active mitochondria. Depolarized mitochondria have decreased membrane potential and cannot hold the dye. In our experiment, at a 200 μM H_2_O_2_ concentration, the MMP reduced to 50% of the untreated cells; hence, this condition was used for further examination. Pre-treatment of cells with CL-EA for 12 h followed by H_2_O_2_ (200 μM) treatment for 2 h displayed a dose-dependent increase in MMP with full recovery at 10 and 25 μg/mL (^###^
*p* < 0.001). Pre-treatment with CL-H also restored MMP to 10 μg/mL (*p* < 0.01), but it decreased afterward (^##^
*p* < 0.01) (Figure 9) which might be due to toxicity. Pre-treating the cells with eugenol, β-caryophyllene, and TMA also resulted in a dose-dependent restoration of MMP. Eugenol at 0.1 μg/mL displayed a significant effect (^#^
*p* < 0.05), and at 1 and 10 μg/mL, it completely reversed the effect of H_2_O_2_ (^###^
*p* <0.001) and established MMP as in the untreated cells. β-caryophyllene and TMA shared the same trend and displayed highly significant recovery (^###^
*p* <0.001) at 1 and 10 μg/mL for β-caryophyllene, and 0.1–10 μg/mL for TMA (Figure 9).

### 2.10. Pre-Treatment of Clove Extract and Bioactive Compounds Restored Oxidative Stress Markers altered by H_2_O_2_-Induced Oxidative Stress

Next, we studied the effect of clove extract, eugenol, β-caryophyllene, and TMA on the H_2_O_2_-induced oxidative stress-affected parameters (GSH, GSSG, and MDA) in the SH-SY5Y cells. 

#### 2.10.1. Restoration of Glutathione Levels

As shown in Figure 10, pre-treatment of SH-SY5Y cells with varying concentrations of clove extract (1, 10, and 30 μg/mL) for 24 h before 6 h incubation with H_2_O_2_ (100 μM) resulted in a dose-dependent increase in GSH level, where CL-EA displayed a significant increase (^#^
*p* < 0.05) at 10 μg/mL and a complete recovery at 30 μg/mL (^#^
*p* < 0.001). On the other hand, CL-H also increased GSH levels to 30 μg/mL, though the effect was non-significant. Additionally, among the three pure compounds, only β-caryophyllene (0.1, 1, and 10 μg/mL) showed a slight dose-dependent increase in GSH levels, but that too was non-significant. 

The oxidized glutathione levels were also measured (Figure 11). The H_2_O_2_-induced stress elevated intracellular GSSG levels. In our study, CL-H significantly (^#^
*p* < 0.05) reduced GSSH levels at 1 μg/mL; however, the decrease was non-significant at higher concentrations (10 and 30 μg/mL). Meanwhile, CL-EA displayed a dose-dependently significant (^#^
*p* < 0.05) decrease in GSSG at 30 μg/mL. Eugenol also helped in decreasing the elevated GSSG and displayed a significant (^#^
*p* < 0.05) effect at 0.1 μg/mL, after which the decrease was non-significant. β-Caryophyllene modestly decreased GSSG levels, while on the contrary, TMA restored GSSG levels at 0.1 and 1 μg/mL (^#^
*p* < 0.05). The increased GSSG at 10 μg/mL might be due to toxicity at this concentration.

The GSH/GSSG ratio is an indicator of a cell’s health. In our study, only CL-EA exerted a dose-dependent increase in the GSH/GSSG ratio towards the normal state significantly at 30 μg/mL (^#^
*p* < 0.05). CL-H displayed a significant effect at a lower concentration of 1 μg/mL (^#^
*p* < 0.05), whereas eugenol and TMA had a non-significant moderate effect and no effect of β-Caryophyllene (Figure 12). 

#### 2.10.2. Attenuation of MDA Levels by Clove Extract

MDA arises from the peroxidation of polyunsaturated fatty acids present in the membrane and forms toxic adducts with proteins and nucleic acids. MDA levels increase with stress; hence, they are used as markers of oxidative damage. In the present study, clove extracts decreased MDA content in a dose-dependent manner, though the results were non-significant (Figure 13). The decrease in MDA levels was significant (^#^
*p* < 0.05) in eugenol (1 and 10 μg/mL) and TMA (10 μg/mL). However, β-caryophyllene exhibited the most potent response by reducing levels of MDA significantly (^#^
*p* < 0.05) at all the concentrations tested. 

## 3. Discussion

In addition to being an important spice, *Syzygium aromaticum* has also been used in traditional medicine to cure several diseases. In the present study, the underlying neuroprotective mechanism of *S. aromaticum* (Clove) extract and its major bioactive compounds were studied under hydrogen peroxide-induced oxidative stress in human neuroblastoma SH-SY5Y cell lines as a model. 

In the present study, phytochemical analysis for TPC and TFC, along with antioxidant potential, was carried out. Phenols and flavonoids are secondary metabolites with an important role in growth, communication, and plant defense [19,20]. As per the GC-MS, CL-H has a higher content of eugenol and eugenol acetate, which possess strong antioxidant activities by hydrogen/electron transfer or directly trapping the free radicals [45]. β-Caryophyllene, a sesquiterpene detected in GC-MS of Clove extract, is also a moderate antioxidant [46,47]. Likewise, the better antioxidant potential of CL-H compared to CL-EA observed in the antioxidant assays can be correlated to the higher phenolic content in CL-H. A positive correlation between phenolic content and antioxidant potential has already been reported [21]. 

The involvement of AGEs in the pathology of NDDs is evident [48]. The AGEs are generated from the irreversible non-enzymatic glycation (Maillard reaction) between proteins and reducing sugars [49], which leads to the formation of cross-linked β structures that eventually form amyloid fibrils, causing protein dysfunction/enzyme deactivation. The binding of AGEs to RAGE (receptors of AGEs) recruits cellular signaling, triggering nuclear factors and leading to the increased expression of inflammatory cytokines and oxidative stress. As observed in our study, Eugenol exhibited anti-AGE activity (IC_50_ 152 μg/mL), while β-caryophyllene and TMA failed to display significant inhibition. Based on the structure-activity relationship (SAR) study, the presence of a hydroxyl/nitro group and halogen atoms in the phenol ring enhances antiglycation activity; on the other hand, alkyl, trifluoromethyl, and bulky groups reduce the activity. Likewise, the activity is lost by phenyl group replacement by hetaryl or nonaromatic groups or if a blend of various scaffoldings is present in a single molecule [50]. Additionally, methoxy groups decrease the inhibition [51], as observed for TMA. Eugenol is an allyl-substituted guaiacol-derived phenylpropanoid. It has a phenolic hydroxyl group, a methoxy group, and an allylic double bond. Whereas β-caryophyllene is a bicyclic sesquiterpene having a cyclobutene ring and a *trans*-double bond in a 9-membered ring. Hence, due to structural complexity and the absence of a hydroxy group in β-caryophyllene, it is unable to display significant anti-glycation activity. Furthermore, the Clove reduced protein carbonyl content and thiol oxidation by scavenging hydroxy and superoxide radicals generated during oxidative reactions (auto-oxidation of sugar, degradation of Amadori products) in the AGE formation [34]. The higher anti-glycation potential of CL-EA compared to a single metabolite, eugenol, suggests the synergistic effect of other compounds present in the extract for anti-glycation action [35]. Since eugenol and β-caryophyllene are common in both extracts, CL-EA showed better inhibition than CL-H, indicating the synergistic action of the phytocompounds exclusively present in CL-EA in boosting anti-AGE activity. Additionally, higher flavonoid content in CL-EA can also result in higher glycation inhibition, as a strong correlation between the two has already been reported [52]. Consequently, CL-EA can be considered a new potential antiglycation agent of natural origin with reduced toxicity for the development of anti-AGE drugs in the management of NDDs. 

We have also investigated the anti-AChE potential and mode of inhibition by Clove extracts and their components in an in vitro system. The extracts and components displayed potential inhibition against AChE (electric eel) with a competitive mode of inhibition. Better IC_50_ values of pure compounds compared to the extract can be explained based on the presence of various metabolites in the extract, which might have diluted the effect. The electric eel AChE has Trp_286_, Phe_297_, Tyr_337_, Tyr_341_, and His_447_ at the active site [53], and β-caryophyllene has been shown to interact with Phe_297_ and Trp_286_ [54] at the active site. The anti-AChE activity of eugenol derivatives was found to be dependent on substitution at the hydroxyl group, as the p-substituted ethyl group had better activity than the methyl derivative [55]. These results suggest that both β-caryophyllene and eugenol bind to the active site of the AChE enzyme as competitive inhibitors. As indicated by the kinetic parameters, Vmax (the maximal reaction velocity when the enzyme is saturated with its substrate) remains unchanged, but Km (the concentration of the substrate that permits the enzyme to achieve half Vmax) increases from normal, indicating a competitive inhibition. Km is also an indicator of the binding affinity for the enzyme (the lower the Km, the higher the affinity). As the competitive inhibitors can bind to E and not to the ES complex (hence cannot change the catalysis in ES), the Vmax is unaffected. The apparent Km for the substrate is increased in the presence of competitive inhibitors, as a higher concentration of substrate will be required to overcome the inhibitory effect. In the present study, CL-EA exhibited the lowest Km value, followed by eugenol, CL-H, and caryophyllene, indicating weak binding and hence less competition for the enzyme. 

Under normal conditions, Aβ is present in soluble form in cerebrospinal fluid (CSF) and blood. However, in AD pathology, it is fibrillated as the key component of amyloid plaques. Aβ has been known to boost free radical production, resulting in oxidative stress and neuroinflammation, ultimately leading to cognitive and behavioral decline [56]. The π-stacking interactions between aromatic side chains of Aβ are responsible for amyloid aggregation [57]. Eugenol has been reported to suppress individual aggregation as well as coaggregation and exert a dual role in preventing amyloid formation by stabilizing the native protein conformation and delaying fibril formation [56]. Additionally, a compound (2′-Hydroxy-4′-methoxyacetophenone) similar to TMA alleviated Aβ deposition in APP/S1 mice [57]. The hydroxyl and methoxy groups in the molecule are involved in multiple non-covalent bindings with globular proteins [58]. Additionally, the 4′-hydroxyl group of eugenol specifically interacts with the lysine residue of the protein [33], which might prevent/delay fibrillization. Eugenol can easily cross the BBB and is c apable of reducing the hippocampal Aβ plaques in animal models [59,60]. Therefore, it is speculated that aromatic compounds in CL-EA and CL-H interfere with β sheet structure of the protein through π-stacking or hydrophobic interaction, impairing Aβ fibrillization. The lower inhibitory activity displayed by eugenol and β-caryophyllene compared to the extract in our study suggests the synergistic or additive effects of phytocompounds present in clove extracts for the anti-amyloidogenic property.

Oxidative stress is one of the important factors in the pathogenesis of AD. Overproduction of ROS induces oxidative stress, which further hampers normal cellular activity by damaging proteins/enzymes and other cellular components, including mitochondria. Therefore, it is important to identify compounds that can reduce intracellular ROS. The dose-dependent reduction in ROS observed in our study indicates the antioxidant nature of the extracts and compounds tested. Both eugenol and β-caryophyllene are the main components of clove extract, which are well-known for their antioxidant action at low concentrations by preventing lipid peroxidation and preserving the levels of antioxidative enzymes [58,59,60,61,62,63] via maintaining oxidative balance through the sirtuin-1 (SIRT1) pathway [62]. Antioxidant activity is increased by the presence of the methoxy group and the number of hydroxyl groups in the phenol ring [59]. Eugenol scavenges free radicals by donating the phenolic hydroxyl group [23]. Much literature is unavailable on TMA, but Paeonol (2′-Hydroxy-4′-methoxyacetophenone), a structurally similar compound to TMA, has shown neuroprotection in vivo and in vitro by reducing oxidative stress [64,65]. A previous study suggested that eugenol, being lipophilic, influences the redistribution of protons across the membrane, thereby affecting membrane potential [66]. Aspirin eugenol ester exerted a shielding effect on oxidative stress generated on H_2_O_2_-induced PC12 cells by scavenging free radicals, reducing intracellular ROS, stabilizing mitochondrial membranes, and increasing the ΔΨm by regulating the PI3K/Akt signal pathway [67,68]. β-caryophyllene has also been reported to inhibit ROS production and recovery of MMP via Cannabinoid type 2 (CB2) receptor-dependent nuclear factor erythroid 2–related factor 2 (Nrf2) activation [69,70]. Likewise, a close relative of TMA, Paeonol, attenuated oxidative stress and mitochondrial dysfunction in streptozotocin (STZ) mice in a model of sporadic Alzheimer’s disease (sAD) [71].

GSH is a potent endogenous antioxidant that protects cells from oxidative stress. These results indicate that CL-EA is most effective in restoring levels of GSH that were dropped due to oxidative stress induced by H_2_O_2_ treatment. Even though individual compounds were not able to increase GSH levels on their own, it can be suggested that the higher efficiency of CL-EA is the result of the synergistic action of the bioactive components. GSH restoration by CL-EA is supported by earlier findings where flavonoids protected the cells against oxidative stress by upregulating the GSH-redox system [72]. The improved GSH/GSSG ratio is an indicator of the effective conversion of GSSG to GSH catalyzed by glutathione reductase (GR) in the presence of clove extract, especially CL-EA, and is supported by a previous study on the recovery of neuronal oxidative stress by GSH/GSSG conditioning in the presence of an antioxidant [73]. Eugenol and eugenol acetate inhibit lipid peroxidation by donating their phenolic hydrogen atoms to trap peroxyl radicals that cause lipid peroxidation [74,75]. Hence, it can be concluded that clove extract can reduce lipid peroxidation due to the synergistic effects of the key bioactives present. Our results are also supported by a previous finding that clove extract (hydroalcoholic) improved antioxidant status in animal studies [76]. 

To summarize, the extracts have a variety of components in different concentrations that have an impact on various cellular pathways either by synergistic/additive or antagonistic effects. CL-H is showing better anti-AChE activity, anti-fibrilization potential, better neuroprotection, and ROS scavenging, whereas CL-EA displayed a superior profile in Aβ oligomerization and fibrilization, anti-glycation, and restoring MMP, glutathione, and MDA levels. The difference in action can be explained based on the concentration of phytocompounds present in them, the mechanism of action, and the affected pathways. 

## 4. Materials and Methods

### 4.1. Chemicals

Acetylcholinesterase (*Electrophorus electricus*, Type VI-S), 6,6′-dinitro-3,3′-dithiodibenzoic acid, bis(3-carboxy-4-nitrophenyl) disulfide (DTNB), acetyl thiocholine chloride, galantamine, gallic acid, ascorbic acid, 2,2′-azinobis-(3-ethylbenzothiazoline-6-sulfonic acid) (ABTS), 2,2-diphenyl-1-picrylhydrazyl (DPPH), 2,4,6-tripyridyl-s-triazine (TPTZ), Folin–Ciocalteu reagent (FCR), eugenol, β-caryophyllene, 2′,3′,4′-trimethoxy acetophenone, 2′,7′-dichlorofluorescin diacetate (DCFDA), tetramethylrhodamine, ethyl ester (TMRE), hydrogen peroxide, thioflavin T (ThT), bovine serum albumin (BSA), sodium azide, aminoguanidine, dextrose, thiobarbituric acid (TBA), trichloroacetic acid (TCA), malondialdehyde (MDA), oxidized glutathione (GSSG), reduced glutathione (GSH), O-phthaldehyde (OPT), sodium hydroxide, N-ethylmaleimide (NEM), Triton X-100 were purchased from Sigma-Aldrich (St. Louis, MO, USA). The BCA protein estimation kit was from Thermo Scientific (Waltham, MA, USA). Aβ_1–42_ (Aggresure™) was procured from AnaSpec (Fremont, CA, USA). The WST-8 kit was purchased from Roche Diagnostics GmbH (Mannheim, Germany). Aβ_1–42_ for MDS was obtained from GenicBio Inc. (Shanghai, China). Phosphate Buffered Saline (PBS) containing 0.05% Tween, and 3,3′,5,5′-Tetramethylbenzidine solution (TMB) were purchased from Thermo Fisher Scientific (Waltham, MA, USA). The purified anti-β-Amyloid, 1–16 antibody, and the horseradish peroxidase (HRP)-conjugated W0-2 monoclonal antibody was purchased each from Biolegend (San Diego, CA, USA) and Peoplebio Inc. (Seongnam, Republic of Korea). All organic solvents of HPLC grade were purchased from Sigma-Aldrich (St. Louis, MO, USA). 

### 4.2. Plant Material and Extraction

The dried clove buds were purchased from the Expat Mart in Seoul, Republic of Korea. The pre-weighted samples were powdered using a pestle and mortar. The powder was extracted successively using n-hexane and ethyl acetate. The extracted fractions were dried, weighed, and kept at 4 °C until further experiments. 

### 4.3. Gas Chromatography-Mass Spectrometry (GC-MS) Method

The sample was separated on a fused-silica capillary column (DB-5ms UI, 30 m × 0.25 mm i.d., film thickness 0.25 μm, Agilent, Santa Clara, CA, USA) installed on GCMS-QP2020 (Shimadzu, Kyoto, Japan). The oven temperature was programmed as isothermic at 60 °C for 2 min, 100 °C at 4 °C/min, 290 °C at 10 °C/min, and finally isothermic for 10 min. The split injection mode (1:10) was used. The carrier gas was helium at a constant flow rate of 1 mL/min. The injection port, ion source, and interface temperatures were 280, 280, and 150 °C, respectively. The energy of ionization was 70 eV. The mass spectra were obtained in full scan mode (40-700 AMU). The hexane and ethyl acetate fractions (1 μL, 1 mg/mL) were injected into the GC/MS via an auto-injector. The unknown compounds were identified by matching known compounds in the National Institute of Standards and Technology (NIST) library.

### 4.4. Determination of Total Phenolic Content

The total phenolic concentration in the extracts was determined as per the Folin−Ciocalteu method [77], with modifications for a 96-well format. The extract was incubated with Folin−Ciocalteu reagent (1N) for 5 min at room temperature (RT), followed by the addition of sodium carbonate (100 g/L in deionized water). The plate was incubated in the dark for 2 h at RT, and the absorbance was measured at 765 nm using a microplate reader (Synergy-H1 BioTek, Agilent, Santa Clara, CA, USA). Gallic acid (10–200 mg/L) was used as a standard for calibration. The experiment was carried out in triplicate, and the results are expressed as mg gallic acid equivalent (GAE)/g of extract.

### 4.5. Determination of Total Flavonoids Content

The total flavonoids were estimated using a previously described method [78], with modifications. To the extract, 10% aluminum chloride and 150 µL of 96% ethanol and sodium acetate (1M) were added. The reagents were mixed, and the plate was incubated at RT for 40 min in the dark. The absorbance was measured at 415 nm using a microplate reader (Synergy-H1 BioTek, Agilent, Santa Clara, CA, USA). The standard curve of quercetin (10–100 µg/mL) was used to estimate flavonoids in the extract and was expressed as mg quercetin equivalents per gram of sample (mg/g).

### 4.6. Determination of Antioxidant Capacity

#### 4.6.1. Free Radical Scavenging by 2,2-diphenyl-1-picrylhydrazylhydrate (DPPH) Radical

The DPPH radical scavenging capacity of extracts was determined using a method [79] with minor modifications. The diluted extract was mixed with ethanolic DPPH (120 µM). The plate was incubated in the dark for 30 min at RT, and the absorbance was monitored at 515 nm using a plate reader (Synergy-H1 BioTek, Agilent, Santa Clara, CA, USA). Ascorbic acid was used as a standard (0.1–10 µg/mL). Radical scavenging activity (RSA) was calculated using the following formula:% RSA = [(A_b_ − A_s_/A_b_)] × 100
where A_b_ = Absorbance of the blank and A_s_ = Absorbance of the extract.

#### 4.6.2. Free Radical Scavenging by 2,2′-Azino-bis (3-ethylbenzothiazoline-6-sulfonic Acid) [ABTS] Radical

The free radical scavenging capacity of extracts was measured with minor modifications to the method described earlier [80]. Equal volumes of ABTS (0.7 mM) and potassium persulfate (2.45 mM) were mixed and kept in the dark at RT for 30 min to generate ABTS radicals. Later, to the extract, an ABTS radical solution was added and kept in the dark for 30 min at RT. The absorbance was measured at 734 nm using the microplate reader (Synergy-H1 BioTek, Agilent, Santa Clara, CA, USA). Ascorbic acid (100 µg/mL) was used as a standard. The percentage of inhibition of ABTS^+•^ was calculated as: % RSA = [(A_b_ − A_s_/A_b_)] × 100
where A_b_ = Absorbance of the blank and A_s_ = Absorbance of the extract.

#### 4.6.3. Ferric Reducing Antioxidant Potential (FRAP) Assay

The ability of extracts to chelate metal ions was determined by the FRAP assay, which was carried out by modifying a previously described method [81]. The working FRAP reagent was prepared by mixing 10:1:1 volumes of 300 mM acetate buffer (pH 3.6), 10 mM TPTZ (2,4,6-tri(2-pyridyl)-*s*-triazine) in 40 mM hydrochloric acid, and 20 mM ferric chloride. A standard curve was prepared using FeSO_4_.7H_2_O at various concentrations (1 mM). For the assay, the extract was mixed with 300 µL of FRAP reagent, and the reduction of ferric tripyridyltriazine to a ferrous complex by the extract was monitored at 593 nm (microplate reader, Synergy-H1 BioTek, Agilent, Santa Clara, CA, USA) after 30 min of incubation at RT in the dark. FRAP values were expressed as µM Fe^2+^/g of the sample. All measurements were carried out in triplicate.

### 4.7. Advanced Glycation End-Product (AGE) Inhibition Activity

The bovine serum albumin (BSA) glycation reaction was carried out as described earlier [35] by incubating the extract with BSA (50 mg/mL) in phosphate buffer (100 mM, pH 7.4), dextrose monohydrate (0.5 M), and sodium azide (5 mM) at 37 °C for 14 days. Aminoguanidine was used as a positive control. The fluorescence (Ex 370 nm; Ems 440 nm) was measured (Synergy-H1 BioTek, Agilent, Santa Clara, CA, USA), and the percent inhibition of glycation was calculated as:Inhibition (%) = [(C − T)/C × 100]
where C and T are fluorescence intensity in the absence and presence of the sample, respectively. The IC_50_ values were determined by GraphPad Prism 9.5.

### 4.8. Acetylcholinesterase Inhibitory Activity

AChE activity was monitored with slight modifications to Ellman’s method [82]. The extracts were incubated with AChE (4 ng) and ATCC (10 mM) in phosphate buffer (100 mM, pH 7.6) at 37 °C for 15 min. The reaction was terminated by DTNB (15 mM) and incubated for 10 min. The absorbance was measured at 412 nm using the plate reader (Synergy-H1 BioTek, Agilent, Santa Clara, CA, USA). Galantamine served as a positive control. The percent inhibition was calculated as:Percent Inhibitory activity (I%) = [(A1 − A2) − (B1 − B2)]/(A1 − A2) × 100
where A1 is the absorbance without inhibitor; A2 is the negative control without inhibitor; B1 is the absorbance with inhibitor and B2 is the negative control with inhibitor.

The IC_50_ values were determined by GraphPad Prism 9.5. For inhibition kinetics, substrate concentration was varied (0.1–1.6 mM), keeping other conditions the same as described above. Km and Vmax were calculated from the Lineweaver–Burk plot.

### 4.9. Thioflavin T (ThT) Assay

A previously reported method was followed with slight modification [40]. The assay was performed using 5 μM Aβ_1–42_ (Aggresure™ AnaSpec) in PBS and incubated with or without extract (100 μg/mL) at 37 °C for 24 h. Later, 100 μM ThT was added, and the plate was incubated for 15 min at 37 °C, after which the fluorescence (Ex 450 nm; Ems 490 nm) was measured (Synergy-H1 BioTek, Agilent, Santa Clara, CA, USA). Phenol red (50 μM) served as the inhibitor control. 

The aggregation inhibition was calculated as: Percent Inhibition (%) = [(1 − F_i_/F_c_) × 100%]
where F_i_ and F_c_ are the fluorescence intensity with and without the inhibitors, respectively. 

### 4.10. Multiple Detection System (MDS)

MDS was conducted as in the previous report [83]. In brief, Aβ_1–42_ (GenicBio Inc., Hong Kong, China) was reconstituted in PBS and stored at −80 °C. The samples were collected at 0 h, 2 h, and 4 h during incubation with the extracts and Aβ_1–42_ at RT. The collected samples were stored at −80 °C until further use. 

The clear 96-well plate (Grenier Bio-One, Kremsmünster, Austria) was coated with anti-β-Amyloid, which targeted Aβ, and incubated overnight at 4 °C. After incubation, the plate was washed three times with PBS containing 0.05% Tween (PBST). The 2% bovine serum albumin was added for blocking and incubated at RT for 2 h. The prepared samples were incubated at RT for 1 h after diluting 4 times. The detection solution containing horseradish peroxidase (HRP)-conjugated W0-2 monoclonal antibody was incubated at RT for 30 min. 3,3′,5,5′-Tetramethylbenzidine solution (TMB) was applied as a color-changing reagent and incubated at RT for 15 min. The optical density was measured at 450 nm wavelength using a microplate reader (Victor3, PerkinElmer) after adding a stop solution.

The area under the curve (AUC) was calculated as: AUC = Sum of 0, 2, 4 h values

### 4.11. Cell Culture 

Human neuroblastoma SH-SY5Y cells (ATCC CRL-2266) were maintained in Dulbecco’s modified Eagle’s medium (DMEM, Gibco) supplemented with 10% fetal bovine serum (FBS), 1% Kanamycin, and 1% Penicillin (Thermo Fisher Scientific, Waltham, MA, USA) at 37 °C in a 95% humidified atmosphere containing 5% CO_2_ in the incubator. The cells were passaged twice per week, and the experiments were performed at 80–90% confluency.

#### 4.11.1. Cell Viability Assay

Briefly, 1 × 10^4^ cells were seeded per well in sterile 96-well plates and exposed to different concentrations of extracts for 24 h. The cells were washed twice with PBS (1X) and incubated in the fresh medium with 10% WST-8 reagent (Roche, Germany) for 2 h. The absorbance was measured at 450 nm in a multiplate reader (Synergy-H1 BioTek, Agilent, Santa Clara, CA, USA). The percent cytotoxicity was calculated as:Cytotoxicity% = (A_c_ − A_t_)/(A_c_) × 100
where A_c_ = absorbance of the control cells and A_t_ = Absorbance of the treated cells.

The plot of percent cytotoxicity versus sample concentration was used to calculate the extract concentration that killed 50% of the cells (IC_50_).

#### 4.11.2. Neuroprotective Activity Assay

The neuroprotective activity in H_2_O_2_-induced oxidative stress in SH-SY5Y was performed as previously described [40]. The cells (1 × 10^4^ cells/well) were seeded in a 96-well plate and incubated for 18–24 h. After stabilization, cells were pre-treated with the extracts for 24 h before 6 h of incubation with H_2_O_2_ (100 μM). A solvent control, H_2_O_2_ alone, and extract alone treatments were also included. After incubation, the % of cell viabilities were determined using the WST-8 reagent in triplicate experiments. 

#### 4.11.3. Measurement of Intracellular Reactive Oxygen Species (ROS)

The ROS was measured using 2′,7′-dichlorodihydrofluorescein diacetate (H2DCFDA) as previously described [84]. The cells (1 × 10^4^ cells/well) were seeded in a 96-well plate and incubated for 18–24 h after which they were pre-treated with the extract for 12 h, followed by 4 h of treatment with H_2_O_2_ (100 μM). Later, 25 μM H2DCFDA was added, and the cells were incubated for another 2 h in the dark at 37 °C. Fluorescence intensity (Ex 495 nm, Ems 520 nm) was measured by a microplate reader (Synergy-H1 BioTek, Agilent, Santa Clara, CA, USA). The ROS was calculated as a percentage of the untreated control cells (100%) in triplicate measurements.

#### 4.11.4. Mitochondrial Membrane Potential (ΔΨm) Assay

The mitochondrial membrane potential was measured using the tetramethyl rhodamine, methyl ester (TMRE) staining method [85]. The cells (1 × 10^4^ cells/well) were seeded in a 96-well plate and incubated for 18–24 h after which they were pre-treated with the extract for 12 h, followed by 2 h of treatment with H_2_O_2_ (200 μM). The cells were incubated for 1 h with 1 μM TMRE at 37 °C. The fluorescence (Ex 549 nm, Ems 575 nm) was read in a microplate reader (Synergy-H1 BioTek, Agilent, Santa Clara, CA, USA). The ΔΨm was calculated as a percentage of the untreated control cells (100%) in triplicate measurements.

#### 4.11.5. Antioxidant Parameters in Cell Lysate

The cells (5 × 10^4^ cells/well) were seeded in a 6-well plate and incubated for 18–24 h. After stabilization, cells were pre-treated with the extracts for 24 h before 6 h of incubation with H_2_O_2_ (100 μM). A solvent control, H_2_O_2_ alone, and extract alone treatments were also included. The culture media was removed, and the cells were washed with cold PBS (1X). Placed the cells on ice and incubated with pre-chilled phosphate buffer (100 mM, pH 7.4) containing 1% Triton X-100 for 30 min. Remove the lysate in microfuge tubes and spin at 20,000× *g* for 10 min at 4 °C (Labogene 1730R, BMS, Paju-si, Republic of Korea) [86]. The supernatant was collected and stored at −80 °C.

##### Protein Estimation

The protein concentration in the samples was measured using the BCA protein estimation kit (Thermo Scientific, Waltham, MA, USA). The BSA standard (10–1000 μg/mL) was used to calculate protein concentration in the unknown samples. 

##### Estimation of Glutathione

The concentrations of GSH (reduced) and GSSG (oxidized) in the lysate were measured fluorometrically using a previously described method [87] with modifications [88], and the fluorescence was recorded at 350/420 (Ex/Ems) in a microplate reader (Synergy-H1 BioTek, Agilent, Santa Clara, CA, USA).

##### Estimation of Malondialdehyde (MDA)

MDA levels were measured by modifications to the method of Heath and Packer [89]. In the reaction, MDA reacts with two molecules of thiobarbituric acid (TBA) to give a pink pigment that absorbs at 532 nm. The standard curve of MDA (1–100 μM) was used to calculate lipid peroxidation in the lysate. 

### 4.12. Statistical Analysis

Statistical analysis was established by one-way ANOVA followed by Dunnett’s post hoc test. The data are registered as the mean ± SD of at least three experiments. The symbols ###, *** represent *p <* 0.001; ##, ** represent *p <* 0.01; and #, * represent *p <* 0.05. The symbol # indicates significance compared to H_2_O_2_ control, while * indicates significance compared to untreated control. The IC_50_ values were determined using non-linear regression, and Lineweaver–Burk plots were drawn using linear regression analysis by GraphPad Prism 9.5. 

## 5. Conclusions

Herein, various cell-based and biochemical studies were conducted to evaluate the neuroprotective potential of *Syzygium aromaticum* (Clove) extracts. Oxidative stress was induced in human neuroblastoma SH-SY5Y cell lines using H_2_O_2_ as it is the key ROS generator and mediator of oxidative damage in AD pathogenesis. Oxidative stress hastens abnormal protein aggregation (Aβ, Tau) which in turn initiates an avalanche of reactions leading to microglia activation, neuroinflammation, and neuronal apoptosis. The clove extract not only provided neuroprotection against H_2_O_2_-induced oxidative stress by reducing ROS, restoring mitochondrial membrane potential, increasing GSH, and decreasing lipid peroxidation in the cells, but also displayed anti-acetylcholinesterase activity, anti-glycation potential, and Aβ oligomerization/fibrilization inhibition. Additionally, the neuroprotective effect of 2′,3′,4′-Trimethoxy acetophenone has been reported for the first time in this study. It is speculated that the neuroprotective mechanism involves a synergistic effect of the bioactive components present in the clove extract, and it would be no exaggeration to mention clove as “Nature’s cure for AD”. This multitarget neuroprotective approach displayed by Clove makes it a promising candidate for AD drug development. We tend to proceed with in vivo studies with Clove in the near future.

## Figures and Tables

**Figure 1 ijms-24-08148-f001:**
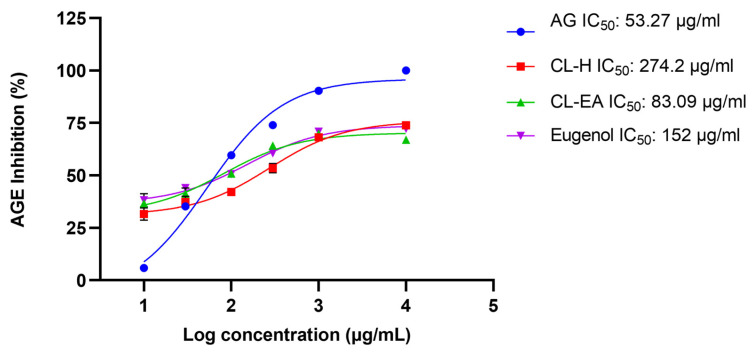
The anti-glycation activity exhibited by Clove extracts and eugenol. IC_50_ values were estimated using the Glucose-BSA model. Data are means of triplicate. Results were calculated using GraphPad Prism 9.5. Abbreviations: AG—Aminoguanidine; CL–H—Clove–Hexane; CL-EA—Clove–Ethyl Acetate.

**Figure 2 ijms-24-08148-f002:**
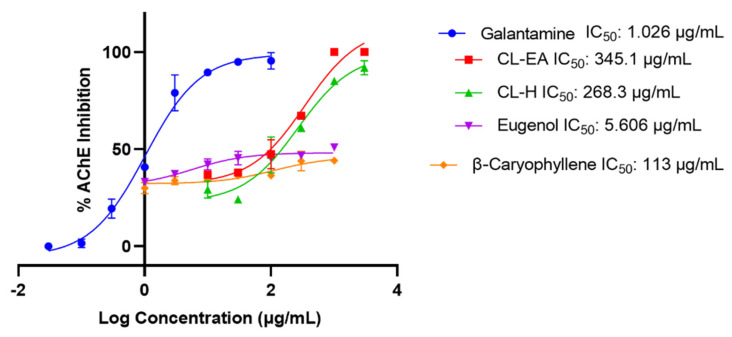
IC_50_ curves of Clove extracts, eugenol, and β-Caryophyllene with inhibitor control (Galantamine) against AChE. The IC_50_ values were calculated using GraphPad Prism 9.5. Abbreviations: CL-H: Clove-Hexane; CL-EA: Clove-Ethyl Acetate.

**Figure 3 ijms-24-08148-f003:**
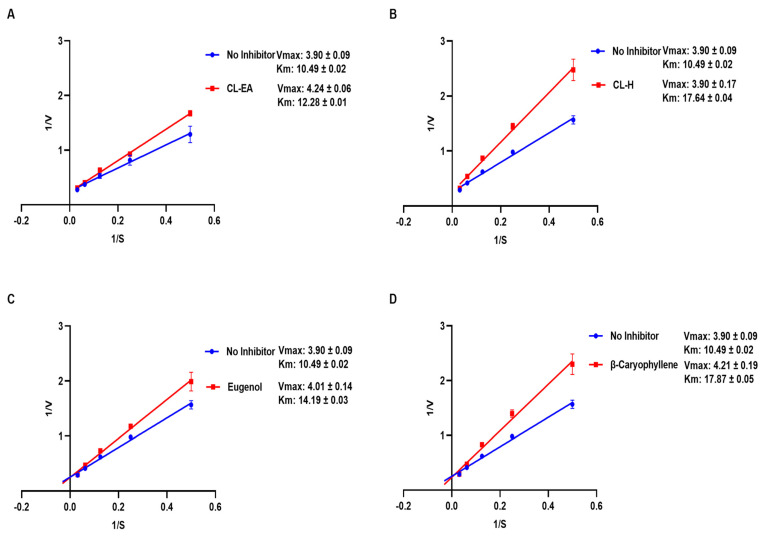
Lineweaver-Burk plot of AChE in the presence of (**A**) CL-EA, (**B**) CL-H, (**C**) eugenol, and (**D**) β-Caryophyllene. Experiments were performed in triplicate. The graph was plotted using GraphPad Prism 9.5. Abbreviations: CL-H: Clove-Hexane; CL-EA: Clove-Ethyl Acetate.

**Figure 4 ijms-24-08148-f004:**
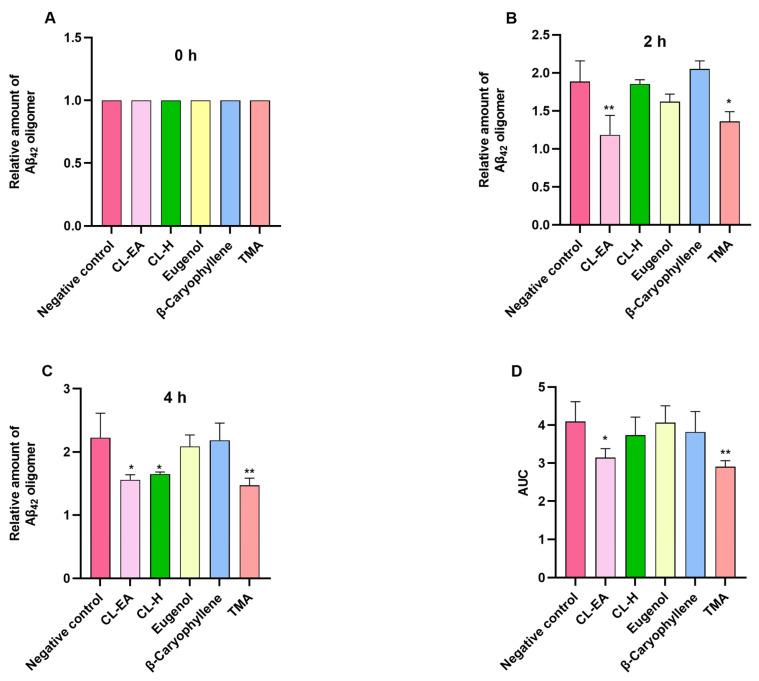
Inhibition effect of Aβ oligomerization by Clove extracts (CL-H and CL-EA), eugenol, β-Caryophyllene, and TMA. The relative amount of Aβ oligomers at 0 (**A**), 2 (**B**), and 4 h (**C**) and the value of Aβ oligomerization area under the curve (**D**). All data are presented as mean ± SEM (*n* = 3). A significant difference * (*p* < 0.05) and ** (*p* < 0.01) using one-way ANOVA followed by Dunnett’s post-hoc was observed in the reduction in oligomerization vs. the negative control (no treatment). Abbreviations: CL-H: Clove-Hexane; CL-EA: Clove-Ethyl Acetate; TMA: 2′,3′,4′-Trimethoxy Acetophenone.

**Figure 5 ijms-24-08148-f005:**
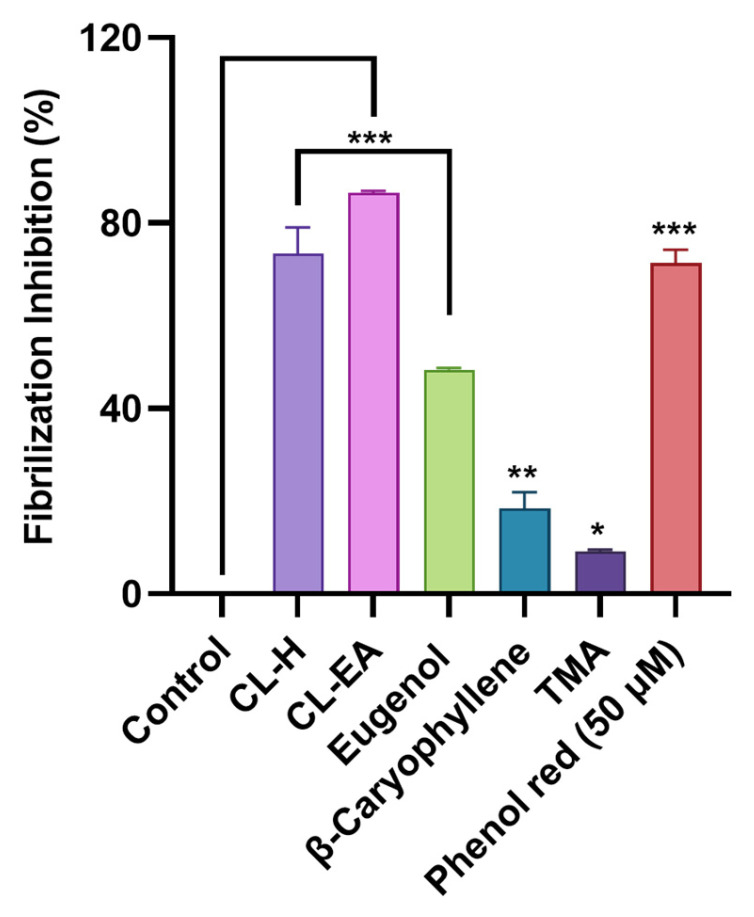
Fibrilization inhibition in the presence of Clove extracts (CL-H and CL-EA), eugenol, β-Caryophyllene, and TMA. The values are expressed as the mean ± SD (*n* = 3). Phenol Red (50 μM) was used as a positive control. A significant difference * (*p* < 0.05), ** (*p* < 0.01), and (*** *p* < 0.001) using one-way ANOVA followed by Dunnett’s post hoc was observed in the reduction in oligomerization vs. the negative control (buffer + Aβ). Abbreviations: CL-H: Clove-Hexane; CL-EA: Clove-Ethyl Acetate; TMA: 2′,3′,4′-Trimethoxy Acetophenone.

**Figure 6 ijms-24-08148-f006:**
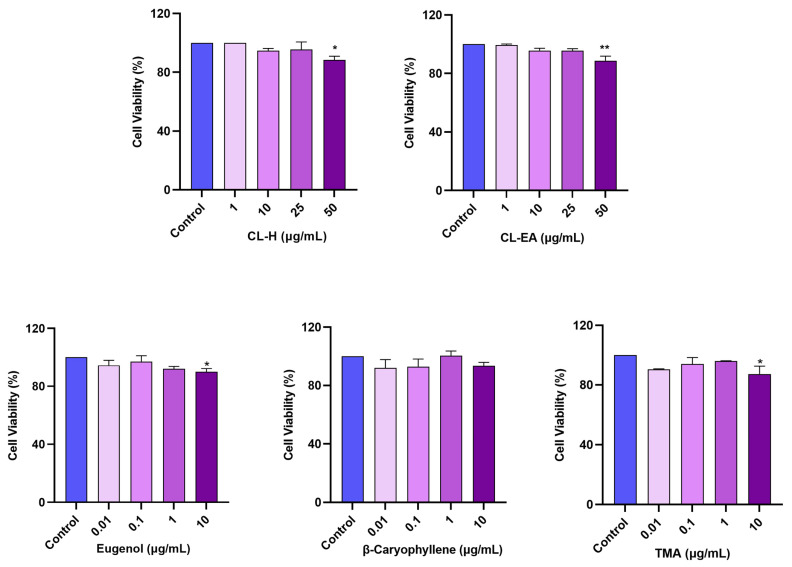
Cytotoxicity assay of Clove extracts (CL-H and CL-EA), eugenol, β-Caryophyllene, and TMA on the SH-SY5Y cells. The cells were treated for 24 h with varying extract concentrations (1, 10, 25, and 50 μg/mL). The cell viability was reported as the percentage of the control group (100%). All data are presented as mean ± SEM (*n* = 3). A significant difference * (*p* < 0.05) and ** (*p* < 0.01) using one-way ANOVA followed by Dunnett’s post hoc was observed in the % of cell viability vs. the control group (no treatment). Abbreviations: CL-H: Clove-Hexane; CL-EA: Clove-Ethyl Acetate; TMA: 2′,3′,4′-Trimethoxy Acetophenone.

**Figure 7 ijms-24-08148-f007:**
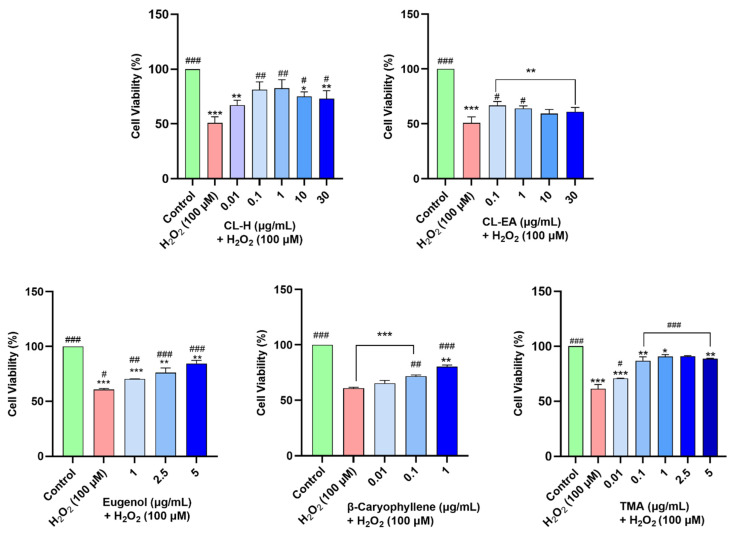
Neuroprotective effects of Clove extracts (CL–H and CL–EA), Eugenol, β-Caryophyllene, and TMA in H_2_O_2_-induced neuroblastoma SH-SY5Y cells. The SH-SY5Y cells were preincubated with the extracts for 12 h, followed by 6 h of H_2_O_2_ (100 μM) treatment. The results indicate % cell viability vs. the control cells mean ± SEM (*n* = 3). A significant difference */^#^ (*p* < 0.05), **/^##^ (*p* < 0.01), and ***/^###^ (*p* < 0.001), using one-way ANOVA followed by Dunnett’s test, was observed in the % of cell viability vs. untreated cells (^*^) and H_2_O_2_ treated cells (^#^). Abbreviations: CL-H: Clove-Hexane; CL-EA: Clove-Ethyl Acetate; TMA: 2′,3′,4′-Trimethoxy Acetophenone.

**Figure 8 ijms-24-08148-f008:**
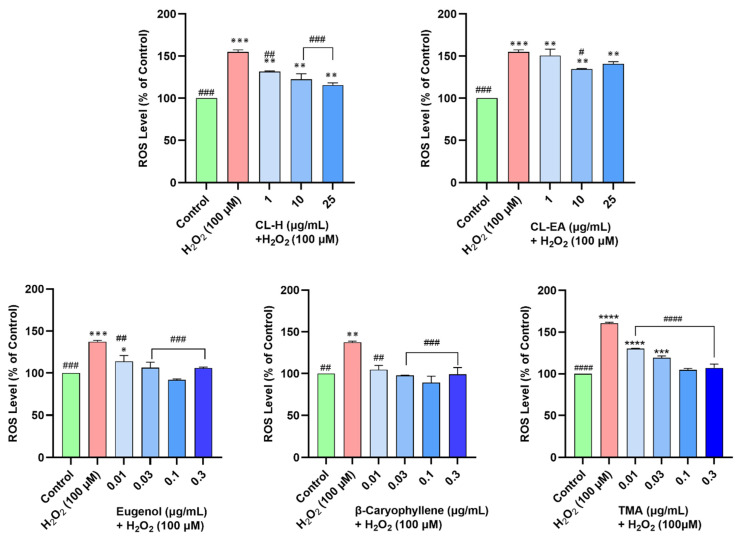
Effect of Clove extracts, eugenol, β-Caryophyllene, and TMA on H_2_O_2_-induced ROS production in SH-SY5Y cells. The SH-SY5Y cells were preincubated with the extracts for 12 h, followed by 6 h of H_2_O_2_ (100 μM) treatment. The results indicate the % of the ROS level vs. the control cells (untreated cells). Values are mean ± SEM (*n* = 3). The data were analyzed by one-way ANOVA followed by Dunnett’s test. A significant difference */^#^ (*p* < 0.05), **/^##^ (*p* < 0.01), ***/^###^ (*p* < 0.001), and ****/^####^ (*p* < 0.0001) was observed in the % ROS vs. untreated cells (^*^) and H_2_O_2_ treated cells (^#^). Abbreviations: CL-H: Clove-Hexane; CL-EA: Clove-Ethyl Acetate; TMA: 2′,3′,4′-Trimethoxy Acetophenone; ROS: Reactive oxygen species.

**Figure 9 ijms-24-08148-f009:**
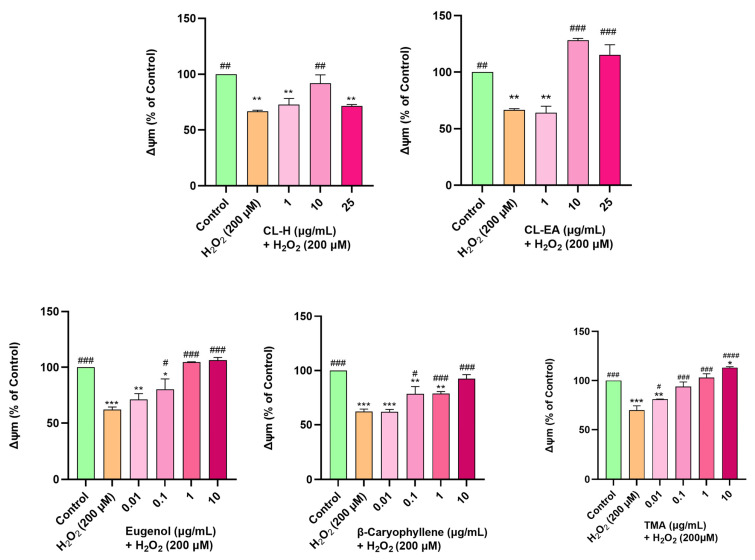
Mitochondrial membrane potential in SH-SY5Y cells exposed to 200 μM H_2_O_2_ for 2 h after 12 h of pre-treatment with Clove extracts, eugenol, β-Caryophyllene, and TMA. The results indicate % ∆Ψm vs. the control cells (untreated cells). Values are mean ± SEM (*n* = 3). The data were analyzed by one-way ANOVA followed by Dunnett’s test. A significant difference */^#^ (*p* < 0.05), **/^##^ (*p* < 0.01), ***/^###^ (*p* < 0.001), and ^####^ (*p* < 0.0001) was observed in the % cell viability vs. untreated cells (^*^) and H_2_O_2_ treated cells (^#^). Abbreviations: CL-H: Clove-Hexane; CL-EA: Clove-Ethyl Acetate; TMA: 2′,3′,4′-Trimethoxy Acetophenone.; ∆Ψm: Mitochondrial membrane potential.

**Figure 10 ijms-24-08148-f010:**
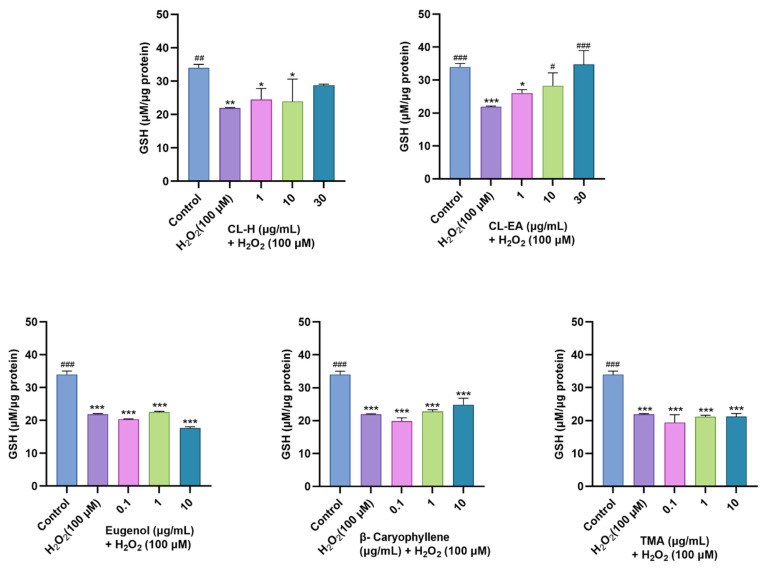
Reduced glutathione content in SH-SY5Y cell lysate exposed to 100 μM H_2_O_2_ for 6 h after 24 h of pre-treatment with Clove extracts, eugenol, β-Caryophyllene, and TMA. The results indicate GSH (μM/μg protein) levels in treated and control cells (untreated cells). Values are mean ± SEM (*n* = 3). The data were analyzed by one-way ANOVA followed by Dunnett’s test. A significant difference */^#^ (*p* < 0.05), **/^##^ (*p* < 0.01), and ***/^###^ (*p* < 0.001) was observed in comparison to untreated cells (^*^) and H_2_O_2_ treated cells (^#^). Abbreviations: CL-H: Clove-Hexane; CL-EA: Clove-Ethyl Acetate; TMA: 2′,3′,4′-Trimethoxy Acetophenone; GSH: reduced glutathione.

**Figure 11 ijms-24-08148-f011:**
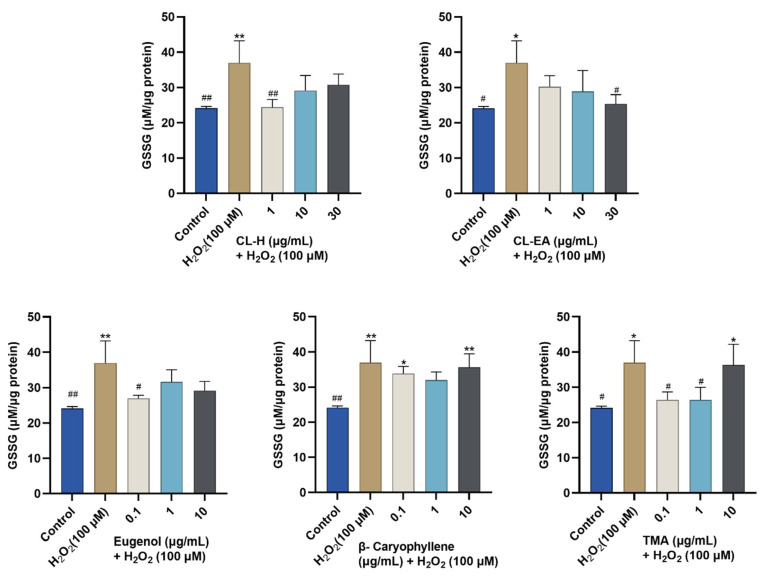
Oxidized glutathione content in the SH-SY5Y cell lysate exposed to 100 μM H_2_O_2_ for 6 h after 24 h of pre-treatment with Clove extracts, eugenol, β-Caryophyllene, and TMA. The results indicate GSSG (μM/μg protein) levels in treated and control cells (untreated cells). Values are mean ± SEM (*n* = 3). The data were analyzed by one-way ANOVA followed by Dunnett’s test. A significant difference of */^#^ (*p* < 0.05), and **/^##^ (*p* < 0.01) was observed in comparison to untreated cells (*) and H_2_O_2-_treated cells (#). Abbreviations: CL-H: Clove-Hexane; CL-EA: Clove-Ethyl Acetate; TMA: 2′,3′,4′-Trimethoxy Acetophenone.; GSSG: oxidized glutathione.

**Figure 12 ijms-24-08148-f012:**
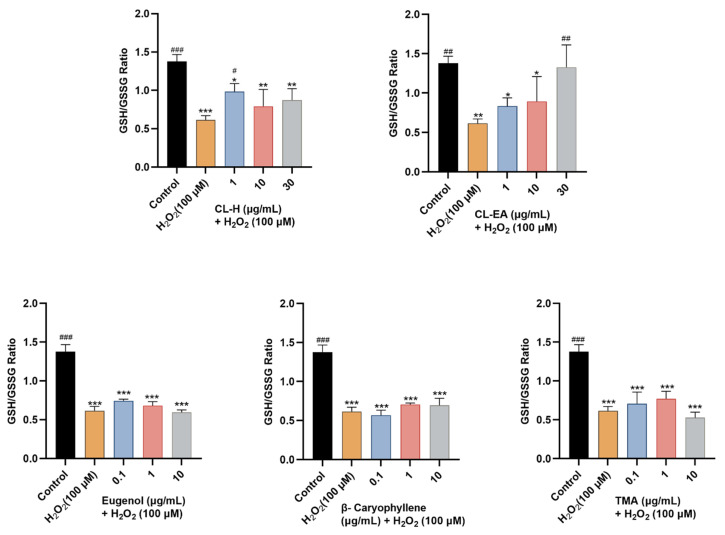
GSH/GSSG ratio in SH-SY5Y cell lysate exposed to 100 μM H_2_O_2_ for 6 h after 24 h of pre-treatment with Clove extracts, eugenol, β-Caryophyllene, and TMA. The results indicate GSH/GSSG ratio in treated and control cells (untreated cells). Values are mean ± SEM (*n* = 3). The data were analyzed by one-way ANOVA followed by Dunnett’s test. A significant difference of */^#^ (*p* < 0.05), **/^##^ (*p* < 0.01), and ***/^###^ (*p* < 0.001) was observed in comparison to untreated cells (*) and H_2_O_2_ treated cells (#). Abbreviations: CL-H: Clove-Hexane; CL-EA: Clove-Ethyl Acetate; TMA: 2′,3′,4′-Trimethoxy Acetophenone; GSH: reduced glutathione; GSSG: oxidized glutathione.

**Figure 13 ijms-24-08148-f013:**
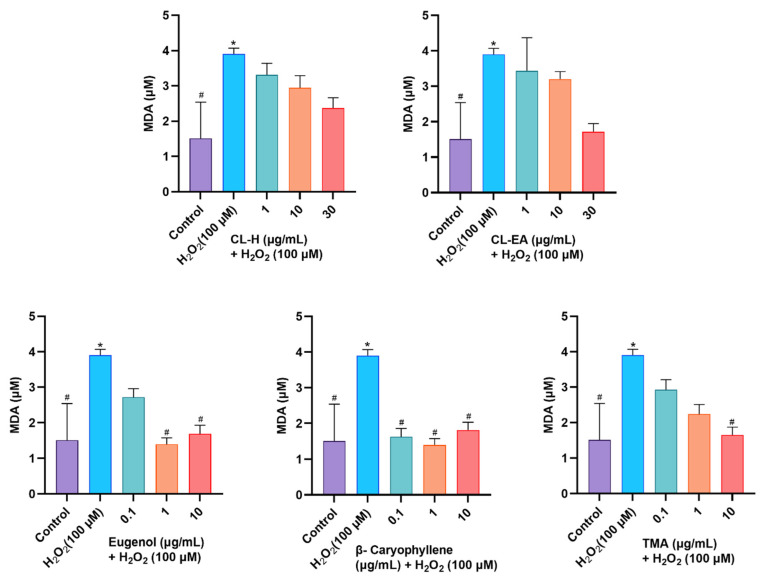
Malondialdehyde (MDA) content in SH-SY5Y cells lysate exposed to 100 μM H_2_O_2_ for 6 h after 24 h pre-treatment with Clove extracts, Eugenol, β-Caryophyllene, and TMA. The results indicate MDA (μM) in treated and control cells (untreated cells). Values are mean ± SEM (*n* = 3). The data were analyzed by One-way ANOVA followed by Dunnett’s test. A significant difference */^#^ (*p* < 0.05 was observed in comparison to untreated cells (^*^) and H_2_O_2-_treated cells (^#^). Abbreviations: CL-H: Clove-Hexane; CL-EA: Clove-Ethyl Acetate; TMA: 2′,3′,4′-Trimethoxy Acetophenone; MDA: Malondialdehyde.

**Table 1 ijms-24-08148-t001:** Kinetic parameters.

	Vmax (μmole/min/mg)	Km (mM)	Type of Inhibition
No Inhibitor	3.90 ± 0.09	10.49 ± 0.02	
CL-H	3.90 ±0.17	17.64 ± 0.04	Competitive
CL-EA	4.24 ± 0.06	12.28 ±0.01	Competitive
Eugenol	4.01 ± 0.14	14.19 ± 0.03	Competitive
β-Caryophyllene	4.21 ± 0.19	17.87 ± 0.05	Competitive

## Data Availability

Data are contained within the article. Additional information is provided in Appendix A.

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
