# Peer review of "Multi-Targeting Neuroprotective Effects of Syzygium aromaticum Bud Extracts and Their Key Phytocompounds against Neurodegenerative Diseases"

_ijms, 2023, doi:10.3390/ijms24098148_

Round 1

Reviewer 1 Report

Find below my considerations about the manuscript:

The title unequivocally values the results obtained. The investigated "t Anti-Alzheimer's" mechanisms are actually secondary mechanisms that are attributed to fighting the disease. Therefore, the title should be completely changed to a title capable of conveying the results obtained by the authors.

The Figure with the chemical compounds must be transformed and inserted in the form of a table. The graphics that accompany this figure should be added as supplemental material.

Figure 2 should match the above graphic.

A specimen number and the name of the botanist responsible for identifying the species must be entered in section 2.2. The exact location with coordinates must be entered in this section.

The gc-ms methodology must be inserted after 2.2.

In addition, it is better for authors to create a workflow for a better overview of the methodologies used.

Was there a standard deviation for Figure 6A?

The authors performed an adequate report of the results of the antioxidant capacity, however, it is necessary to correlate these results in a more interesting way with the chemical constituents found in the gc-ms.

The authors used as a reference and cited papers 60, 61, and 62. In the reference Silvi Ikawati et al the authors used a non-human AChE. Do you know how to inform the identity and similarity of this AChE with the h-AChE? In addition, this paper has some critical points in its methodology. I consider it more appropriate that this part of the text be removed.

Minor editing of English language required

Author Response

Response to Comments

First reviewer 

We wish to express our sincere thanks to the reviewer for the input and suggestions on our manuscript. We have incorporated the changes in the manuscript in red fonts.

Comments and Suggestions for Authors

  1. The title unequivocally values the results obtained. The investigated "t Anti-Alzheimer's" mechanisms are actually secondary mechanisms that are attributed to fighting the disease. Therefore, the title should be completely changed to a title capable of conveying the results obtained by the authors.

√ The title has been changed as per the suggestion.

  1. The Figure with the chemical compounds must be transformed and inserted in the form of a table. The graphics that accompany this figure should be added as supplemental material. Figure 2 should match the above graphic.

√ The changes have been made as suggested by the reviewer and added as supplementary material.

  1. A specimen number and the name of the botanist responsible for identifying the species must be entered in section 2.2. The exact location with coordinates must be entered in this section.

√ We have purchased the packed Cloves from Expat Mart, South Korea and being a marketed product, no voucher specimen is required.

  1. The gc-ms methodology must be inserted after 2.2. In addition, it is better for authors to create a workflow for a better overview of the methodologies used.

√ The changes have been made as suggested by the reviewer and added in the Material & method section (New section #4.3)

  1. Was there a standard deviation for Figure 6A?

√ In MDS results, we used relative value according to the samples at zero hour. That is why all the samples were 1.0 in 6A (New Fig.# 4)

  1. The authors performed an adequate report of the results of the antioxidant capacity, however, it is necessary to correlate these results in a more interesting way with the chemical constituents found in the gc-ms.

√ The manuscript has been revised thoroughly and the changes have been made in the “Discussion” as per the suggestion.

  1. The authors used as a reference and cited papers 60, 61, and 62. In the reference Silvi Ikawati et al the authors used a non-human AChE. Do you know how to inform the identity and similarity of this AChE with the h-AChE? In addition, this paper has some critical points in its methodology. I consider it more appropriate that this part of the text be removed.

√ We have revised the manuscript and removed the references as suggested.

  1. Minor editing of English language required.

√ We have thoroughly revised the manuscript for grammatical corrections as suggested.

Reviewer 2 Report

Article of great interest in the search for effective and safe treatments for Alzheimer's disease.

As indicated, it is important to carry out in vivo studies. For this reason, as it is an ex vivo study, I recommend a change in the title to another that makes it clearer that is an ex vivo study.

Although it has an updated bibliography, it is recommended to include references such as:

Amir Rawa et al. Roles of Syzygium in anti-cholinesterase, antidiabetic, anti-inflammatory, and antioxidant from alzheimer’s oersoective. Plants 2022, 11, 1476.

Shekhar et al. Neuroprotection by ethanolic extract of syzygium aromaticum in Alzheimer’s disease like patology via maintaining oxidative balance trough SIRT1 pathway. Experimental Gerontology. 110 (2018) 277-283

Author Response

Response to Comments

Second Reviewer

Our sincere thanks to the reviewer for the comments and suggestions on the improvement of our manuscript. All changes in the manuscript are indicated in red fonts. 

Comments and Suggestions for Authors

  1. Article of great interest in the search for effective and safe treatments for Alzheimer's disease. As indicated, it is important to carry out in vivo For this reason, as it is an ex vivostudy, I recommend a change in the title to another that makes it clearer that is an ex vivo study.

√ The title has been changed as per the suggestion.

  1. Although it has an updated bibliography, it is recommended to include references such as:

Amir Rawa et al. Roles of Syzygium in anti-cholinesterase, antidiabetic, anti-inflammatory, and antioxidant from alzheimer’s oersoective. Plants 2022, 11, 1476.

Shekhar et al. Neuroprotection by ethanolic extract of syzygium aromaticum in Alzheimer’s disease like patology via maintaining oxidative balance through SIRT1 pathway. Experimental Gerontology. 110 (2018) 277-283

√ The references have been cited as suggested by the reviewer.

Reviewer 3 Report

Evident Anti-Alzheimer’s properties of Syzygium aromaticum bud Extract and its Key Bio actives in SH-SY5Y Neuroblastoma Cells and Biochemical studies 

The premise:

In the present study, the underlying neuroprotective mechanism 

of an important culinary spice Syzygium aromaticum (Clove) extract, and major bioactive compounds 

were studied in hydrogen peroxide-induced oxidative stress in human neuroblastoma SH-SY5Y cell 

lines as a model.  The extract protects against ROS, and mitochondrial stress and restored GSH content in addition to anticholinergic, anti-glycation and AB aggregation inhibition.

The abstract is informative and balanced.

Introduction:

The authors summarise the key pathological aspects of AD well in the first paragraph.

The next paragraph talks about various therapies and the associated side effects and the need to develop new therapies.

SA extract is then introduced and its key properties get mentioned and this is then linked to ROS production in NDD. Having built momentum, then the authors set out their key objectives. Excellent.

Methods and materials

Detailed isolation, extraction and quantification methods.

Method 2.5-9: the authors tested non-cell-based methods for antioxidant, free radical, free reducing antioxidant, acetylcholinesterase inhibitory capacity

Then cell culture and cell-based assays have been introduced (2.10). Finally, in 2.11 mass-spec is used.

3. Results

3.2. In this section, the authors could explain why the CL-EA (clove ethyl acetate) and CL-H (clove hexane) extracts give different profiles, please add this to the results section here. Also, what impact this can have on the future formulation of these components? In addition, any of the other components identified in the extracts could have NDD protective properties which have not been covered here.

3.3. In Figure 2 legend please kindly elaborate on the structure versus function of the compounds (if known).

In the paragraph that follows the figure legend of Figure 3, there is some context given that may be a little too much detail for the results and more suitable for discussion. I have now reached the end of the paper and it seems that the authors have combined the discussion and results and there is no separate discussion. I would suggest adding a separate discussion.

In the section where Figure 3 is explained please compare the IC50 of the two types of extracts more clearly (I see this appears in the first paragraph of page 11). The authors could mention what the implication of a higher anti-glycation of CL-EA compared to H is for future use and formulation.

The use of line-weaver burke to compare competitive inhibition is useful, please add the Vm and Km values to the Figure 5 legend or figure as well since it currently lacks orientation. From Table 1, the lowest Km is demonstrated by CL-EA (suggesting high affinity), please explain this in the text in comparison to other Kms.

In Figure 6 please use in-figure annotations to better distinguish the four plots (for example show that time is the factor; A-C). In the first few lines of page 14, you have missed the Figure 6A label. As the time of incubation progressed, CL-H also decreased oligomerization significantly (*p <0.05) like CL-EA however more significant reduction was shown by TMA (**p <0.01) at 4h (Figure 6C). Please explain in the text why. Could the compound get degraded with time? With respect to the AUC, it seems CL-EA is performing better, what is the implication?

Figure 7, convincing data.

I realise that NB cell lines get used frequently for AD studies, however, these are cancer cells, how about differentiating them with RA/BDNF and retesting them or using neural progenitor cells such as ReNcell to test a more physiological state (in AD the objective is to keep the cells alive and preventing their apoptosis, while the opposite is aimed in cancer studies). In Figure 8, only higher concentrations of the extracted lead to reduced cell viability and this is probably due to increased toxicity and side effects at any rate.

In general, the authors consistently use their controls which is reassuring.

Figure 9, an interesting trend in that the two extracts continue to reduce cell viability at high concentrations, while in Figure 10 this trend is seen for ROS levels, isn’t this contradictory? Please explain the reason. The trend for the controls in 9 compared to 10 makes sense though.

Figure 11, 

Pre-treatment of cells with CL-EA for 12 h followed by H2O2 (200 μM) treatment for 2 h displayed a dose-dependent increase in MMP with full recovery at 10 and 25 μg/mL (###p <0.001). Pre-treatment with CL-H also restored MMP at 10 μg/mL (p <0.01), but it decreased afterwards (##p <0.01) (Figure 11).  The result for the two extracts is interesting in that the potential increases but then drops with the highest dose. You also show that the highest dose reduced viability so this may be linked. Could the authors elaborate on this?

In Figure 12, the comment I made above for Figure 11, does not apply since the highest does still restores the CSH level how can the authors explain this even though viability is reduced? 

The picture is a lot muddied with Figure 13. Why is the effect for CL-H all over the place but gets reduced for CL-EA? The authors need to explain this and the implication for formulation. This is also showcased in Figure 14 since CL-EA improves the index. Why? Please explain in the text.

In general, not enough justification for the results has been provided, giving context is one thing but trying to explain your results is another.

The authors need to add a separate section for discussion and mention the justification for many of the trends, limitations of the experiment and the implication and finally which of the two extracts, in balance, is more beneficial for AD. They could move the discussion parts from the individual results section and consolidate it into a main discussion, so there really won’t be much remodelling required.

Competent English needs only a little proofreading and checks.

Author Response

Response to Comments

Third reviewer

Comments and Suggestions for Authors

We are grateful to the reviewer for his valuable suggestions on our manuscript. All changes in the manuscript are indicated in red fonts.

  1. In this section, the authors could explain why the CL-EA (clove ethyl acetate) and CL-H (clove hexane) extracts give different profiles, please add this to the results section here. Also, what impact this can have on the future formulation of these components? In addition, any of the other components identified in the extracts could have NDD protective properties which have not been covered here.

√ As per the suggestions, we have revised the “Results” (Section 2.2) and included the desired information.

  1. In Figure 2 legend please kindly elaborate on the structure versus function of the compounds (if known).

√ This figure has been moved to the supplementary section.

  1. In the paragraph that follows the figure legend of Figure 3, there is some context given that may be a little too much detail for the results and more suitable for discussion. I have now reached the end of the paper and it seems that the authors have combined the discussion and results and there is no separate discussion. I would suggest adding a separate discussion.

As per the reviewer’s suggestion a separate “Discussion” (section 3) has been included in the manuscript.

  1. In the section where Figure 3 is explained please compare the IC50 of the two types of extracts more clearly (I see this appears in the first paragraph of page 11). The authors could mention what the implication of a higher anti-glycation of CL-EA compared to H is for future use and formulation.

The information has been included in the “Discussion” (section 3) as suggested.

  1. The use of line-weaver burke to compare competitive inhibition is useful, please add the Vm and Km values to the Figure 5 legend or figure as well since it currently lacks orientation. From Table 1, the lowest Km is demonstrated by CL-EA (suggesting high affinity), please explain this in the text in comparison to other Kms.

√ As per the reviewer’s suggestion, the information has been incorporated in the “Discussion” (section 3).

  1. In Figure 6 please use in-figure annotations to better distinguish the four plots (for example show that time is the factor; A-C). In the first few lines of page 14, you have missed the Figure 6A label. As the time of incubation progressed, CL-H also decreased oligomerization significantly (*p <0.05) like CL-EA however more significant reduction was shown by TMA (**p <0.01) at 4h (Figure 6C). Please explain in the text why. Could the compound get degraded with time? With respect to the AUC, it seems CL-EA is performing better, what is the implication?

√ The figure 6 (New Fig# 4) has been annotated and the information has been included in the section 2.5 as suggested.

  1. I realise that NB cell lines get used frequently for AD studies, however, these are cancer cells, how about differentiating them with RA/BDNF and retesting them or using neural progenitor cells such as ReNcell to test a more physiological state (in AD the objective is to keep the cells alive and preventing their apoptosis, while the opposite is aimed in cancer studies). In Figure 8, only higher concentrations of the extracted lead to reduced cell viability and this is probably due to increased toxicity and side effects at any rate. In general, the authors consistently use their controls which is reassuring.

√ We are thankful to the reviewer for sharing the idea of using differentiation the cell lines/using ReNcell. We would like to plan this study in future.

8.Figure 9, an interesting trend in that the two extracts continue to reduce cell viability at high concentrations, while in Figure 10 this trend is seen for ROS levels, isn’t this contradictory? Please explain the reason. The trend for the controls in 9 compared to 10 makes sense though.

√ We agree with the reviewer’s comments that the low concentrations of extracts were able to provide better neuroprotection (increase cell viability) compared to the higher concentration (Fig. 7, Previous Fig #9). In Fig 8 (Previous Fig#10), ROS is decreased in a concentration-dependent manner. It implies the role of multiple pathways that affect ROS generation.

  1. Figure 11, Pre-treatment of cells with CL-EA for 12 h followed by H2O2 (200 μM) treatment for 2 h displayed a dose-dependent increase in MMP with full recovery at 10 and 25 μg/mL (###p <0.001). Pre-treatment with CL-H also restored MMP at 10 μg/mL (p <0.01), but it decreased afterwards (##p <0.01) (Figure 11).  The result for the two extracts is interesting in that the potential increases but then drops with the highest dose. You also show that the highest dose reduced viability so this may be linked. Could the authors elaborate on this?

√ In CL-H, MMP decreased significantly at 25 μg/mL which could be due to the toxic effect of CL-H (25 μg/mL) on the mitochondria resulting in decreasing mitochondrial membrane potential.

  1. In Figure 12, the comment I made above for Figure 11, does not apply since the highest does still restores the GSH level how can the authors explain this even though viability is reduced? 

√ The extracts have a variety of components in different concentrations which have an impact on various cellular pathways either by synergistic/additive or antagonistic effect. CL-H is showing better neuroprotection and ROS scavenging whereas CL-EA displayed a better profile in restoring MMP, glutathione, and MDA levels. The difference in action can be explained based on the concentration of phytocompounds present in them and the pathways affected.

  1. The picture is a lot muddied with Figure 13. Why is the effect for CL-H all over the place but gets reduced for CL-EA? The authors need to explain this and the implication for formulation. This is also showcased in Figure 14 since CL-EA improves the index. Why? Please explain in the text.

√ The same reason as explained above can be given in this context also. Additionally, there is an inverse relation between reduced and oxidized glutathione. In stress conditions, GSH is decreased and GSSG is increased, as GSH is oxidized to GSSG (Previous Fig# 13-14).

  1. In general, not enough justification for the results has been provided, giving context is one thing but trying to explain your results is another.

√ We have thoroughly revised the manuscript and included “Discussion” separately with detailed explanation.

13.The authors need to add a separate section for discussion and mention the justification for many of the trends, limitations of the experiment and the implication and finally which of the two extracts, in balance, is more beneficial for AD. They could move the discussion parts from the individual results section and consolidate it into a main discussion, so there really won’t be much remodelling required.

√ As per the reviewer’s suggestion, a separate “Discussion” (section 3) has been added.

Round 2

Reviewer 1 Report

After the modifications made, the manuscript is ready for publication in this journal.

Reviewer 3 Report

The authors have addressed my comments.